# Bivalvular Endocarditis Due to Polymicrobial Coinfection with *Enterococcus faecalis* and *Coxiella burnetii*: A Case Report and Review of the Literature

**DOI:** 10.3390/medicina60071122

**Published:** 2024-07-11

**Authors:** Asala Abu-Ahmad, Fadel Bahouth, Mirit Hershman-Sarafov, Alona Paz, Majed Odeh

**Affiliations:** 1Infectious Diseases Unit, Bnai Zion Medical Center, Faculty of Medicine, Technion-Israel Institute of Technology, Haifa 3104802, Israel; asala.abu-ahmad@b-zion.org.il (A.A.-A.); mirit.heshman@b-zion.org.il (M.H.-S.); alona.paz@b-zion.org.il (A.P.); 2Department of Cardiology, Bnai Zion Medical Center, Faculty of Medicine, Technion-Israel Institute of Technology, Haifa 3104802, Israel; fadelbahouth@gmail.com; 3Department of Internal Medicine A, Bnai Zion Medical Center, Faculty of Medicine, Technion-Israel Institute of Technology, Haifa 3104802, Israel

**Keywords:** endocarditis, polymicrobial, bivalvular, Q fever, *Enterococcus faecalis*, *Coxiella burnetii*

## Abstract

Polymicrobial endocarditis is uncommon, and polymicrobial endocarditis in combination with *Coxiella burnetii* is very rare. We herein describe an extremely rare case of polymicrobial bivalvular endocarditis due to coinfection with *Enterococcus faecalis* and *Coxiella burnetii* in a 62-year-old male patient, and extensively review the relevant medical literature. To the best of our knowledge, only three similar cases have been previously reported. Q fever is a worldwide endemic bacterial zoonosis, but it and its most common chronic complication, endocarditis, are still underestimated and underdiagnosed worldwide. This situation reflects the paucity of reported cases of polymicrobial endocarditis in combination with *Coxiella burnetii*. Clinical presentation of Q fever endocarditis is highly nonspecific, and diagnosis may be delayed or missed, leading to severe and potentially fatal disease. Our case and the previously reported similar cases emphasize the need for further evaluation of infective endocarditis due to *Coxiella burnetii*, in all cases of culture-negative endocarditis, and in prolonged oligo-symptomatic inflammatory syndrome, particularly in the presence of valvular heart disease. This approach should be applied even when typical pathogens are isolated, especially in endemic areas of Q fever, and with atypical presentation.

## 1. Introduction

Polymicrobial endocarditis is quite uncommon, with an incidence ranging from 1% to 6.8%, as reported in the medical literature [1,2,3,4]. It occurs mainly in intravenous drug abusers (IVDAs), patients with prolonged intravenous infusion, co-morbidities, congenital heart disease, and patients with prosthetic valves. Reported organisms include mainly *Staphylococcus aureus*, coagulase-negative *Staphylococci*, *Enterococci*, Gram-negative bacilli, *Streptococcus viridans*, anaerobes, and fungi. Mortality and morbidity rates are high [1,2,3,4]. Endocarditis due to coinfection with *Enterococcus faecalis* and *Coxiella burnetii* is extremely rare, and to the best of our knowledge, only three cases of endocarditis due to coinfection with these two bacteria have been previously reported in the medical literature [5,6,7]. In one of these three cases, the endocarditis was bivalvular [6].

We describe a patient with polymicrobial bivalvular endocarditis caused by coinfection with *Enterococcus faecalis* and *Coxiella burnetii* and review the relevant medical literature.

## 2. Case Presentation

A 62-year-old male was admitted to our hospital for a 10-day history of diffuse abdominal pain, and 9 kg weight loss for the last month. He denied fever, chills, nausea, vomiting, diarrhea, or night sweats. His other medical history included hypertension, diabetes mellitus type 2, peripheral arterial disease, recurrent biliary pancreatitis, and laparoscopic cholecystectomy for cholecystitis due to biliary stones. He had no history of endocarditis, valvular heart disease (several months prior to admission, he had a normal echocardiogram), or intravenous drug abuse.

Physical examination revealed a blood pressure of 140/85 mmHg, heart rate of 90 bpm, and normal body temperature. Cardiac auscultation revealed a systolic murmur at the apex, graded 2/6, and the lungs were clear to auscultation. Abdominal examination revealed diffuse mild tenderness, normal peristalsis, and no organomegaly. Other physical findings were unremarkable.

Blood tests demonstrated mild normocytic anemia (12.30 g/dL), normal leukocyte count (8610/mm^3^) with 80% neutrophils, and mild thrombocytopenia (132,000/mm^3^). C-reactive protein was elevated (103 mg/L—normal value < 6 mg/L), liver enzymes were mildly elevated, bilirubin, amylase, lipase, and renal function tests were within normal ranges, and urinalysis was normal. An abdominal computerized tomography (CT) scan with intravenous contrast revealed a wedge-shaped parenchymal defect in the inferior aspect of the left kidney (Figure 1A), and a smaller wedge-shaped parenchymal defect in the spleen (Figure 1B). These two findings were highly suspicious for infarcts. Five out of six blood culture sets obtained on admission were positive for ampicillin-sensitive *Enterococcus faecalis.*

A trans-thoracic echocardiogram (TTE) revealed normal left ventricular (LV) and right ventricular (RV) function, moderate mitral regurgitation, and a 0.9 × 0.8 cm mobile mass in the atrial aspect of the posterolateral leaflet of the mitral valve, as well as a small mass on the aortic valve with normal function of the valve, consistent with vegetation on both valves. Trans-esophageal echocardiogram revealed the same findings (Figure 2). The diagnosis of infective endocarditis of the mitral and aortic valve was established.

The patient started treatment with ampicillin (12 g/day) and ceftriaxone (4 g/day). He was also investigated for culture-negative endocarditis, and serological tests returned positive for chronic Q fever (Table 1). Q fever serology, as reported in Table 1, was performed in the Israeli reference lab (Institute for Biological Research, Ness-Ziona, Israel). The reference standard test for the serologic diagnosis of Q fever is the indirect fluorescent antibody (IFA) test using *Coxiella burnetii* antigen with two-fold dilutions of serum from 1:100. Cut-off depends on the IFA technique used, whether in-house developed or commercial. Definite serologic diagnosis confirming chronic Q fever is based on IgG titers. IgM titers may support a diagnosis of acute Q fever. Thus, our patient had negative IgM titers, as expected in chronic Q fever infection. The patient lives in Haifa city in northern Israel with his wife. He denied any recent history of fever, cough, or headache, exposure to domestic animals, ingestion of unpasteurized milk and dairy products, and history of tick bites.

In the presence of bivalvular vegetation, spleen and kidney infarcts, and positive serology for chronic Q fever, a diagnosis of Q fever endocarditis was made, in addition to the diagnosis of Enterococcal endocarditis. Oral doxycycline (200 mg/day) and hydroxychloroquine (600 mg/day) were added to the treatment regimen. On the twelfth day of admission, the patient was discharged after clinical and laboratory improvement and repeated negative blood cultures for outpatient therapy. He was recommended to complete a 6-week treatment course of intravenous ampicillin and ceftriaxone, and to continue oral doxycycline and hydroxychloroquine for at least 18 months.

One month later, the patient was examined in our outpatient infectious diseases clinic. He was in a good clinical condition without fever, chills, weight loss, effort dyspnea, or signs of heart failure. He completed six weeks of the antibiotic regimen for enterococcal endocarditis and is currently continuing doxycycline and hydroxychloroquine with good adherence and tolerance. Repeated TEE showed normal LV and RV function, moderate mitral regurgitation, small soft mass (0.5 × 0.5 cm) on the atrial aspect of the posterolateral leaflet of the mitral valve, and normal aortic valve with no vegetation (Figure 3).

Repeated Q fever serology showed no further elevation in phase I titers (Table 1). The patient underwent a positron emission tomography CT (PET-CT) scan, which was interpreted as normal without any enhancement (Figure 4). Repeated third TEE 2 months after the termination of 18 months of antibiotic treatment of *Coxiella burnetii* demonstrated mild mitral regurgitation and no vegetation on the mitral and the aortic valves.

## 3. Discussion

We herein described an extremely rare case of a patient with polymicrobial bivalvular endocarditis due to coinfection with *Enterococcus faecalis* and *Coxiella burnetii*. Infective endocarditis is a serious life-threatening clinical condition occurring in 3–9 cases per 100,000 person-years in developed countries, with a high morbidity and mortality rate. In most cases, it is caused by a single pathogen, including, most commonly, *Staphylococci*, *Streptococci*, and *Enterococci* [8,9,10,11,12,13]. Cases of polymicrobial endocarditis are rare, with an incidence ranging from 1% in unselected populations to 6.8% in IVDAs of all cases of infective endocarditis [1,2,3,4]. The mortality rate of polymicrobial endocarditis is higher than that of monomicrobial endocarditis and is associated with much more cardiac surgery for controlling the infection, or for repairing damaged valves, particularly in IVDAs [13,14,15,16,17,18]. In our patient, the combination of the microorganisms responsible of his polymicrobial endocarditis is extremely rare, and it includes *Enterococcus faecalis* and *Coxiella burnetii*.

Enterococci are the third most common infectious agent of infective endocarditis worldwide, after *Staphylococci* and *Streptococci*, responsible for 9–17% of all cases of infective endocarditis [8,9,10,11,12,13,19], where *Enterococcus faecalis* causes about 90% of all cases [12,19,20,21,22,23,24]. *Enterococcal* endocarditis occurs mainly in the setting of prior valvular damage or prosthetic valve (around 85% of the cases), in elderly male patients who are debilitated, and especially those with gastrointestinal neoplasia. Healthcare association is common [21,22,23,24,25,26,27,28,29,30,31,32,33,34,35,36,37]. Except for being elderly and male, our patient did not have any of these risk factors. *Enterococci* are also commonly involved in polymicrobial endocarditis and are considered the third most common etiology of polymicrobial endocarditis, after *Staphylococci* and *Streptococci* [17,18,21,29]. Among the *Enterococcal* species, as in mono-microbial endocarditis, *Enterococcus faecalis* is also involved in most of the cases of polymicrobial endocarditis, most commonly in combination with *Staphylococcus aureus* [17,18,21,29,30].

Q fever is a worldwide bacterial zoonosis, endemic in almost every country in the world except New Zealand, caused by *Coxiella burnetii*, an obligate intracellular Gram-negative bacterium. Cattle, sheep, goats, cats, rabbits, pigeons, and dogs are the main reservoirs. The bacterium is secreted in the urine, milk, feces, and birth products of contaminated animals. Mainly, this infection and its outbreaks occur in rural areas, often in humans having close contact with infected animals and their products. Contaminated products form aerosols that may spread several kilometers via wind, resulting in Q fever occurrence even in urban areas by inhalation of these contaminated aerosols [31,32,33]. Our patient lived in an urban area in the northern part of Israel and lacked the above traditional risk factors.

Q fever encompasses two broad clinical syndromes: acute and chronic infection. About 60% of individuals are asymptomatic during *Coxiella burnetii* primary infection [32,34]. The symptoms of acute Q fever are nonspecific, and common among them are fever, extreme fatigue, severe headache that is frequently retro-orbital, and abdominal pain. These symptoms may commonly be accompanied by pneumonia and/or hepatitis. Other complications, such as endocarditis, pericarditis, and myocarditis, are rare with acute Q fever. The acute infection is commonly self-limited [31,32,33]. Around 1−5% of acute cases develop to chronic form, which most frequently (around 78% of all cases) is manifested as endocarditis [33,34,35]. Main risk factors associated with progression to endocarditis after primary infection are underlying valvular heart disease, immunosuppression, and genetic predisposition [32,33,34,35,36]. The clinical presentation of chronic Q fever endocarditis is nonspecific: fever is frequently absent or low grade, and the patients can present symptoms such as isolated relapsing fever, chills, weight loss, night sweats, and hepatosplenomegaly [32,33,34,35,36]. Our patient presented with weight loss and diffuse abdominal pain. Valvular vegetations are usually small: the aortic valve is involved in about half, and the mitral valve is involved in about one-quarter of all cases [32,33,34,37]. In our patient, both valves were involved. These patients are often ill for more than one year before the diagnosis is made. The diagnosis of chronic Q fever endocarditis may be delayed or missed, and without prompt diagnosis and appropriate antimicrobial therapy, the course of chronic Q fever is severe and potentially fatal [32,33,34,37,38].

Q fever is endemic in Israel and was first reported in 1949 [39]. However, it is still underdiagnosed and under-reported [39,40], as is the situation worldwide [32,33,41]. *Coxiella burnetii* is responsible for 1–5% of all infective endocarditis cases worldwide [11,32,42,43]; however, its prevalence is underestimated [32,41]. This worldwide underdiagnosis and underestimation may reflect the paucity of reported polymicrobial endocarditis cases in combination with *Coxiella burnetii*. To the best of our knowledge, only 16 cases of polymicrobial endocarditis due to chronic Q fever in combination with other bacteria have been previously reported [5,6,7,42,44,45,46,47,48,49]. In three cases, the coinfection was with *Enterococcus faecalis* [5,6,7], as in the present case. Twelve of these sixteen patients needed valve replacement surgery [5,6,7,42,45,46,47,48], one died of septic shock before performing valve replacement surgery [44], and one after performing replacement surgery [47].

Regarding the previously reported three cases of endocarditis due to coinfection with *Enterococcus faecalis* and *Coxiella burnetii*, Mora-Rillo et al. [5] described a male patient with mitral bio-prosthetic valve infective endocarditis due to *Enterococcus faecalis*. Serology for other causes of infective endocarditis was performed and was positive for chronic Q fever. The patient was treated with a dual-antibiotic regimen for both pathogens, along with valve replacement surgery. Valve samples were positive for both *Enterococcus faecalis* and *Coxiella burnetii*. Rovery et al. [6] described an end-stage renal failure on hemodialysis in a male patient with mitral and tricuspid valve infective endocarditis due to *Enterococcus faecalis*. The patient had no history of previous valve disease. Serology for culture-negative endocarditis was performed and was positive for chronic Q fever. He was treated with a dual-antibiotic regimen for both pathogens, along with tricuspid valve replacement. The valve sample was negative for *Coxiella burnetii*. Yahav et al. [7] described a male patient from Israel with aortic mechanical prosthetic valve endocarditis and aortic root abscess due to *Enterococcus faecalis*. Serology for Q fever as part of a fever of unknown origin investigation was taken and was positive for chronic Q fever. The patient was treated with a dual-antibiotic regimen for both pathogens, along with valve replacement surgery. Valve samples were positive for both *Enterococcus faecalis* and *Coxiella burnetii*. In all three cases, as in our case, the patients were male and elderly, which are risk factors for both *Enterococcus faecalis* endocarditis [21,22,23,25] and *Coxiella burnetii* endocarditis [31,32,34]. In two [5,7] of the three cases, the patients had a prosthetic valve, which represents the most important risk factor for endocarditis due to both pathogens [21,22,23,25,31,32,34]. In all three cases, the patients underwent valve replacement surgery, which is very common in polymicrobial endocarditis [10,15]. Related to our patient, indication for surgery is the presence of large (>10 mm) hypermobile vegetation, particularly with prior systemic embolus and significant valve dysfunction [8,9]. Of these, our patient had only systemic embolus, which alone is not indicative for surgery and, fortunately, he improved significantly without the need for this procedure.

In our patient, besides being polymicrobial, the endocarditis was also multi-valvular, involving the aortic and the mitral valve. Multi-valve infective endocarditis is not rare and represents a considerable proportion of overall cases of infective endocarditis, where it accounts for about 15% of all endocarditis cases. Most commonly, it is double-valve, while triple-valve and quadruple-valve endocarditis are rare [1,4,12,13,28,38,50,51,52,53,54,55,56]. As in our case, overall in the previously reported cases, involvement of the aortic and the mitral valve was the most common combination, with a rate of around 90% [1,4,12,13,28,38,50,51,52,53,54,55,56]. Involvement of multiple valves and obstructive septic embolism is often characteristic of *Staphylococcus aureus* etiology. The most common site of embolism for left-sided infective endocarditis is the brain and spleen, while for right-sided it is the lungs [8,9]. As in single-valve infective endocarditis, *Enterococci* are the third most common infectious agent of multi-valve endocarditis, after *Staphylococci* and *Streptococci*, responsible for 9–17% of all cases [12,51,53]. Among the infectious agents of all multi-valve endocarditis cases, *Coxiella burnetii* is not a common one. As in our case, involvement of the aortic and the mitral valve is the most common combination in *Coxiella burnetii* multi-valve endocarditis [38,50,54,57,58]. In contrast to our patient’s indolent course, multi-valve endocarditis is usually associated with a significantly higher heart valvular surgery rate, relapse rate, and mortality rate than in single-valve endocarditis [51,53,56].

In the previously reported three cases of *Enterococcus faecalis* and *Coxiella burnetii* endocarditis [5,6,7], the diagnosis of chronic Q fever endocarditis was made incidentally by serology as part of culture-negative endocarditis or a fever of unknown origin investigation. The diagnosis was confirmed by the surgical valvular sample in two of them [5,7]. In our patient, the diagnosis of chronic Q fever endocarditis was made only by serology in the absence of indication for valvular surgery. However, we believe that *Coxiella burnetii* had a major role in the pathogenesis of his disease for the following reasons. First, the atypical presentation of infective endocarditis: abdominal pain, absence of fever, chills, or night sweats, and the presence of thrombocytopenia. Second, predisposing heart conditions are observed in around 85% of patients with *Enterococcal* endocarditis [24,26,27]. Our patient had a normal echocardiogram without any valvulopathy several months prior to admission. Therefore, we hypothesize that our patient suffered from chronic Q fever endocarditis that caused valvular damage as a predisposing factor for *Enterococcal* endocarditis. This hypothesis is in agreement with that of other investigators [5,46,47,49,59].

## 4. Conclusions

Q fever is a worldwide endemic bacterial zoonosis, but it and its most common chronic complication, endocarditis, are still underestimated and underdiagnosed worldwide. This situation reflects the paucity of reported polymicrobial endocarditis cases in combination with *Coxiella burnetii*. Our case and the previously reported similar cases emphasize the need for further evaluation of infective endocarditis due to *Coxiella burnetii* in all cases of culture-negative endocarditis and in prolonged oligo-symptomatic inflammatory syndrome, particularly in the presence of valvular heart disease. This approach should be applied even when typical pathogens are isolated, especially in endemic areas of Q fever and with atypical cardiac manifestations.

## Figures and Tables

**Figure 1 medicina-60-01122-f001:**
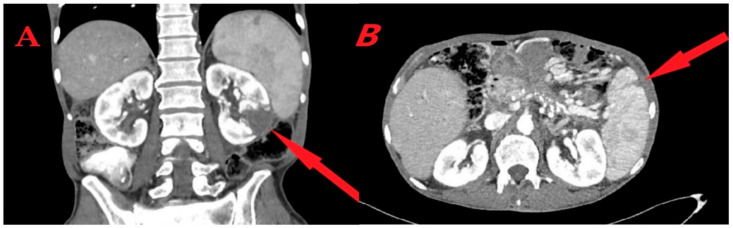
Abdominal CT scan. (**A**) Left kidney infarct—a wedge-shaped parenchymal defect in the inferior aspect of the left kidney (arrow). (**B**) Splenic infarct—a wedge-shaped parenchymal defect in the spleen (arrow).

**Figure 2 medicina-60-01122-f002:**
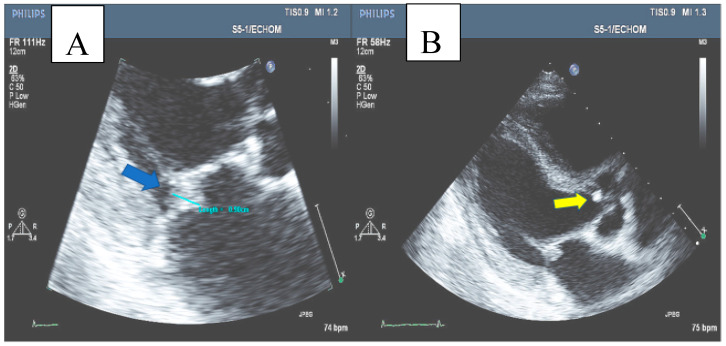
(**A**) Mitral valve vegetation (blue arrow). (**B**) Aortic valve vegetation (yellow arrow).

**Figure 3 medicina-60-01122-f003:**
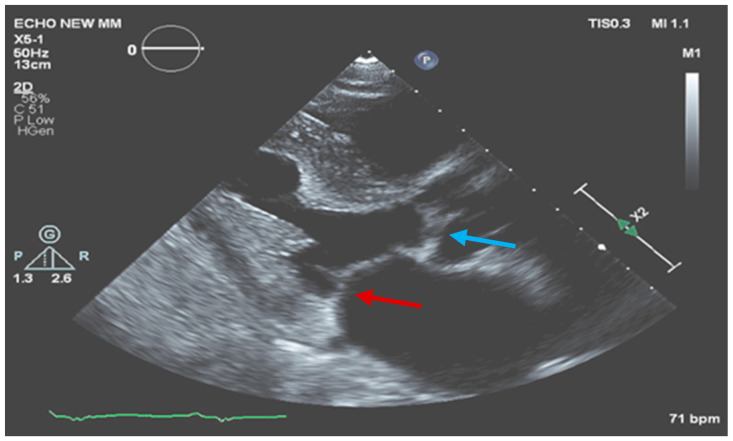
Mitral valve (red arrow) and aortic valve (blue arrow) with significant improvement in the size of the vegetations.

**Figure 4 medicina-60-01122-f004:**
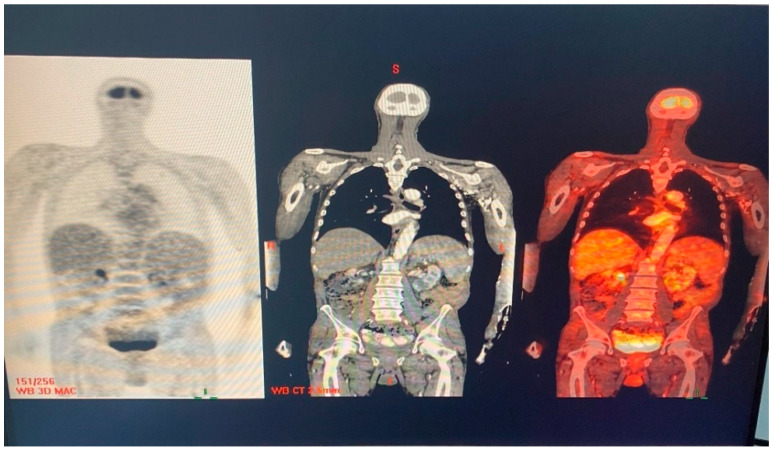
PET-CT scan showing no pathologic enhancement.

**Table 1 medicina-60-01122-t001:** Q fever serology.

	Phase I IgG	Phase I IgM	Phase II IgG	Phase II IgM
**On admission**	1600	Negative	3200	Negative
**2 weeks later**	3200	Negative	1600	Negative
**6 weeks later**	3200	Negative	3200	Negative
**5 months later**	3200	Negative	3200	Negative
**18 months later**	3200	Negative	3200	Negative

## Data Availability

Data are available upon request to the corresponding author.

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
