# Peer review of "Bivalvular Endocarditis Due to Polymicrobial Coinfection with Enterococcus faecalis and Coxiella burnetii: A Case Report and Review of the Literature"

_medicina, 2024, doi:10.3390/medicina60071122_

Round 1

Reviewer 1 Report

Comments and Suggestions for Authors

The authors do a very nice job of describing this case and provide a good review of the literature related to polymicrobial IE with Q Fever. However I have concerns related to the diagnosis in this case that need additional justification before the manuscript could be considered for publication. 

I am not convinced this patient had Q Fever. The authors describe in detail the reasons why the patient is at low risk for Q Fever exposure. Why did they then obtain Q Fever serologies in this case, particularly given that blood cultures were positive? Q Fever serology results are reported in Table 1, but they are not reported in the usual fashion i.e. 1:16, 1:32, 1:64 and so on. What does a 'negative' result mean for IgM titers? The authors need to provide details about the assay used and describe the normal units for this assay. I also note that the serologies did not decrease with treatment, which adds to my concern that this was a false positive serology result. I would expect that serology titer counts would decline with treatment. 

Comments on the Quality of English Language

The quality of English writing and syntax is reasonable. A number of changes to syntax should be considered but these are superseded by other concerns. 

Reviewer 2 Report

Comments and Suggestions for Authors

1.       The case is interesting. Properly managed diagnosis and treatment. Requires little clarification.

2.       A trans -thoracic echocardiogram (TTE) revealed normal left ventricular (LV) and 71 right ventricular (RV) function, moderate mitral regurgitation, and a 0.9x0.8 cm mobile 72 mass in the atrial aspect of the posterolateral leaflet of the mitral valve, and also a small 73 mass on the aortic valve with normal function of the valve, consistent with vegetation on 74 both valves. Trans-esophageal echocardiogram revealed the same findings (Figure 2). The 75 diagnoses of infective endocarditis of the mitral and aortic valves was established.

Such vegetations on the adjacent mitral and aortic valves are called "kissing". Please add.

3.       In the presence of bivalvular vegetation, spleen, and kidney infarcts, and positive 87 serology for chronic Q fever, Q fever endocarditis was diagnosed, in addition to 88 the diagnosis of Enterococcal endocarditis.

Add that the involvement of multiple valves and obstructive embolism is often characteristic of Staphylococcus aureus etiology.

4.       Please add:

- The most common site of embolism for left-sided IE is the brain and spleen, while pulmonary embolism often occurs when right-sided valves or cardiac leads are involved 

- If the vegetations on the left side are over 10mm and there is peripheral embolism then there is an indication for surgery. The patient was therefore on the borderline of indications

- On the right side, most vegetations are treated conservatively with pharmacology, with large vegetations, AngioVac can be considered and the vegetation sucked out if it is in the atrium. Please consider the literature:

AngioVac: The first in Poland percutaneous solid thrombus aspiration from the right atrium.

Puślecki M, Stefaniak S, Katarzyński S, Klotzka A, Buczkowski P, Ładzińska M, Walczak M, Stanisławiak-Rudowicz J, Perek B, Lesiak M, Jemielity M, Grygier M.Kardiol Pol. 2022;80(1):103-104. doi: 10.33963/KP.a2021.0128. Epub 2021 Oct 13

Round 2

Reviewer 1 Report

Comments and Suggestions for Authors

I thank the authors for their detailed and thoughtful responses. I think the manuscript could be published with some additional edits. I do not think we can say this patient had definite Q Fever. The only evidence of Chronic Q Fever in this case is the serologic testing. All of the other valvular findings can be explained by Enterococcus endocarditis. Enterococcus IE can present in an indolent fashion and one month of symptoms is to be expected. I think the authors need to soften their language to say this was a case of Enterococcus endocarditis with possible associated Q Fever endocarditis. 

An interesting angle in this case that the authors should emphasize is that in a population with a high prevalence of Q Fever, there may be an argument for routine screening of all patients presenting with IE for Q Fever. If there is positive serologies, then co-treatment may be warranted. The authors should provide details of the prevalence of positive Q Fever serologies in this region if available. The authors should also emphasize that an antibody response is not always seen despite a long course of treatment, as they have done in the responses. They should also provide an explanation for why they think the phase II IgG antibody was higher than the phase I IgG antibody (this can be seen from time to time, but the authors should note this is atypical).

Comments on the Quality of English Language

Needs review from a native English speaker before could be published.